# Crosstalk between RNA Metabolism and Cellular Stress Responses during Zika Virus Replication

**DOI:** 10.3390/pathogens9030158

**Published:** 2020-02-25

**Authors:** Aarón Oyarzún-Arrau, Luis Alonso-Palomares, Fernando Valiente-Echeverría, Fabiola Osorio, Ricardo Soto-Rifo

**Affiliations:** 1Molecular and Cellular Virology Laboratory, Virology Program, Institute of Biomedical Sciences, Faculty of Medicine, Universidad de Chile, Santiago 8380453, Chile; aaron.oyarzuna@usach.cl (A.O.-A.); luis_alp88@hotmail.com (L.A.-P.); fvaliente@uchile.cl (F.V.-E.); 2HIV/AIDS Workgroup, Faculty of Medicine, Universidad de Chile, Santiago 8380453, Chile; 3Laboratory of Immunology and Cellular Stress, Immunology Program, Institute of Biomedical Sciences, Faculty of Medicine, Universidad de Chile, Santiago 8380453, Chile; fabiolaosorio@med.uchile.cl

**Keywords:** ZIKV, flaviviruses, RNA metabolism, m^6^A, UPR, cellular stress

## Abstract

Zika virus (ZIKV) is a mosquito-borne virus associated with neurological disorders such as Guillain-Barré syndrome and microcephaly. In humans, ZIKV is able to replicate in cell types from different tissues including placental cells, neurons, and microglia. This intricate virus-cell interaction is accompanied by virally induced changes in the infected cell aimed to promote viral replication as well as cellular responses aimed to counteract or tolerate the virus. Early in the infection, the 11-kb positive-sense RNA genome recruit ribosomes in the cytoplasm and the complex is translocated to the endoplasmic reticulum (ER) for viral protein synthesis. In this process, ZIKV replication is known to induce cellular stress, which triggers both the expression of innate immune genes and the phosphorylation of eukaryotic translation initiation factor 2 (eIF2α), shutting-off host protein synthesis. Remodeling of the ER during ZIKV replication also triggers the unfolded protein response (UPR), which induces changes in the cellular transcriptional landscapes aimed to tolerate infection or trigger apoptosis. Alternatively, ZIKV replication induces changes in the adenosine methylation patterns of specific host mRNAs, which have different consequences in viral replication and cellular fate. In addition, the ZIKV RNA genome undergoes adenosine methylation by the host machinery, which results in the inhibition of viral replication. However, despite these relevant findings, the full scope of these processes to the outcome of infection remains poorly elucidated. This review summarizes relevant aspects of the complex crosstalk between RNA metabolism and cellular stress responses against ZIKV and discusses their possible impact on viral pathogenesis.

## 1. Overview of ZIKV Pathogenesis and ZIKV-Associated Diseases

Zika virus (ZIKV) is a mosquito-borne virus member of the flavivirus genus in the *Flaviviridae* family related to dengue virus (DENV), yellow fever virus (YFV), Japanese encephalitis virus (JEV), and West Nile virus (WNV). ZIKV was first isolated from the Zika forest in Uganda in 1947 [1]. Since its first isolation, only a few cases of the disease were recorded in Asia and Africa. However, three major outbreaks were recently reported in Yap Island (2007), French Polynesia (2013) and the widespread epidemic in the Americas between 2015 and 2016 [2]. In most of the cases, ZIKV infection is asymptomatic or produces mild fever, however, in certain circumstances, the infection has been associated with neurological disorders such as Guillain-Barré syndrome and microcephaly, especially with viral strains isolated from the South American region [3]. In January 2016, the World Health Organization declared ZIKV as a global threat to public health making a call to invest in research and the development of vaccines and treatments aimed to control ZIKV-associated diseases [4]. Although ZIKV infections have substantially declined since 2016, as of July 2019, several countries in the Americas reported cases of ZIKV transmission, with the exception of mainland Chile, Uruguay, and Canada. Moreover, by 2018, evidence indicated that ZIKV strains found in the Americas had spread to Angola and were associated with a cluster of microcephaly cases [4,5].

There are two main lineages of the virus: African and Asian. African strains are usually more virulent and cause high cell mortality in vitro, while Asian lineages can induce chronic neurological disorders prior to cell death [6]. Although the American isolates are closely related to the Asian lineage, they are currently classified as a third lineage since they are the only ones able to induce the development of microcephaly [6,7]. ZIKV can be spread through mosquito bites from Aedes spp. (*Aedes aegypti* and *Aedes albopictus*) [8], sexual intercourse, blood transfusion, and vertical transmission. Following infection, ZIKV induces a short viremia in the blood, which can persist for longer periods in other body fluids such as saliva, semen, and urine. Once the infection has occurred and symptoms have disappeared, the virus is able to persist in testes for up to six months. This may explain the asymptomatic transmission and the difficulty to prevent the disease and control the spread of the virus from humans to humans [9]. Vertical transmission during the first trimester of pregnancy is correlated with a higher risk of ZIKV-associated birth defects, including microcephaly (5% to 10% of cases) [10]. In fact, viral particles and viral RNA can be detected in the brain of microcephaly cases, demonstrating direct infection in the central nervous system (CNS) [11,12]. ZIKV is able to infect a wide range of placental cells, including placental trophoblast, endothelial cells, fibroblasts, and fetal macrophages (Hofbauer cells) [13]. Syncytiotrophoblast, one of the main barriers between mother and fetus, is especially resistant to ZIKV infection [14]. However, recent reports suggest that the virus is capable of altering tight junction expression in this tissue, increasing paracellular permeability thus, allowing viral diffusion towards the fetal milieu [15]. ZIKV is also able to infect blood-retinal barrier cells like retinal endothelial cells, retinal pericytes, and retinal pigmented epithelial cells leading to damage in the retina, optic nerve, and retinal vessels in 20 to 55% of congenital ZIKV infections and, although infrequent, major visual impairment in non-congenital infections [16,17,18]. There is also evidence showing that ZIKV can infect human glomerular cells such as human podocytes, renal glomerular endothelial cells, and mesangial cells [19]. Despite only a couple of clinical cases of ZIKV-associated renal complications that have been observed [20], experimental observations suggest that ZIKV could induce acute kidney injury through Nod-like receptor protein 3 (NLRP3) inflammasome activation and the suppression of BCL-2 expression [21].

Although ZIKV seems to be pantropic, it has a strong tropism for neuronal tissue, in particular astrocytes, microglia, oligodendrocytes, and neurons [22,23,24]. This preferential neurotropism can partially explain why ZIKV is able to induce impairment in brain development and neurological disorders. Moreover, ZIKV infection activates several immune responses that lead to the release of proinflammatory cytokines, increased autotrophic activity, and apoptosis induction, while there are also changes in transcriptomic landscapes and metabolic pathways that contribute to the establishment of microcephaly and other associated neural pathologies [25].

## 2. ZIKV Replication Cycle

Similar to other flaviviruses, ZIKV particles are enveloped and contain a capsid with an icosahedral shape. The viral capsid harbors the 11-kb, non-segmented, positive-stranded RNA genome. The viral genomic RNA (gRNA) possesses an m^7^GTP cap structure at its 5’ end and a structured region instead of a poly-A tail at the 3’ end. The gRNA is translated upon entering the cell to generate a polyprotein that contains three structural proteins at its N-terminal domain; capsid (C), envelope (E), prM/M, and seven nonstructural proteins in its C-terminal domain: NS1, NS2A, NS2B, NS3, NS4A, NS4B, and NS5. While C, E, and prM/M correspond to the structural components of the viral particle, the NS proteins participate in viral replication and evasion of the immune response [26].

The viral replication cycle begins with the attachment of the envelope protein E to tyrosine kinase receptors such as AXL, Tyro3 or TIM1, leading to virion endocytosis by a clathrin-dependent mechanism [27] (Figure 1a). Nevertheless, there is controversy about whether these receptors are indeed essential for viral entry [28]. Endosome acidification leads to membrane fusion and gRNA release to the cytoplasm. gRNA translation starts in the cytosol and the presence of a signal peptide in the polyprotein N-terminal end allows the translocation of the translation machinery to the surface of the endoplasmic reticulum (ER). The polyprotein contains several transmembrane domains in a way that upon translation, the unprocessed viral proteins stay within the ER membrane (M, NS2A NS2B, NS4A, NS4B), facing towards the ER lumen (E, NS1) or the cytoplasm (C). The viral polyprotein is processed to the mature viral peptides by the NS3 protease/helicase as well as by ER-resident proteases. Viral RNA replication is carried out by the viral RNA-dependent RNA polymerase (RdRp) NS5 and the viral helicase NS3 [29]. Viral replication and translation are spatially carried out inside membrane folds or compartments in the ER, while gRNA interaction with the capsid protein would allow the assembly of an infectious viral particle [30] (Figure 1a). The viral genomes are enclosed into nascent virions that assemble and bud into the ER lumen, and traffic through the Golgi complex following the secretory pathway and are matured before they are released from the infected cell [31] (Figure 1a).

## 3. Interactions between ZIKV and the Immune System

ZIKV can infect different cell types of the immune system such as microglia, monocytes, macrophages, and dendritic cells [32]. An important early response to virus infection is the production of interferon type-I (IFN-I), which is critical to eliminate the viral invasion [33,34]. Viruses that belong to the *Flaviviridae* family induce the production of IFN-I as well as proinflammatory cytokines such as IL-1β, TNF-α, and IL-6 that are important for the control of viral replication and virus elimination. In addition, ZIKV and other flaviviruses can upregulate TLR3 expression, which is a relevant pattern recognition receptor (PRR) coupling viral sensing to IFN-I production [32,35].

Viruses containing positive-stranded RNA genomes such as ZIKV perform two main functions during the viral replication process as they are used as mRNA to synthesize viral proteins as well as the viral genome that is incorporated into newly produced viral particles [36,37,38]. Upon ZIKV entry into the host cell, the genome is released in the cytoplasm and is immediately translated to synthesize the viral polyprotein [39]. During different stages of viral replication, several viral components must escape from host immune sensors for successful replication. gRNA and proteins can interact with different molecules of the host cell, modulating its activation and altering the antiviral response. During infections with ZIKV, DENV, WNV, and YFV, the viral proteins NS5, NS2B, and NS3 have been proposed as important regulators of type-I IFN production through interferon regulatory factor 3 (IRF3) [40,41,42]. Moreover, ZIKV replication decreases the activation and nuclear translocation capacity of the Signal Transducer and Activator of Transcription Factor 1 and 2 (STAT1 and STAT2) thus, reducing type-I IFN production [41]. In fact, the NS5 protein from ZIKV inhibits human STAT2 favoring viral proliferation and suppressing IFN-I production. Intriguingly, this process is not replicated with murine STAT2 and, in turn, mouse models show a strong type-I IFN response upon ZIKV infection, highlighting relevant differences between mice and human infection outcomes [41] (Figure 1a). It was recently reported that ZIKV infection in mouse macrophages and glial cells induced the inhibition of NLRP3 (NOD-like receptor—NLR) activation, which is important for the inflammasome activation and IL-1β and IL-18 maturation, in a process exerted by the ZIKV NS3 protein [43] (Figure 1a). Other molecules related to IFN production are the small membrane-associated interferon-inducible transmembrane proteins (IFITMs) such as IFITM1 and IFTM3 whose activation has been shown to inhibit ZIKV replication at early stages in HeLa cells [44]. There is a high complexity in the innate immune regulation by the viral proteins with some data resulting contradictory when dissecting details of the virus-host response, which can be explained by the use of different viruses from different lineages as well as different infection models either in vitro and in vivo amongst different research groups (Figure 1a).

On the other hand, ZIKV NS5, NS4B, and NS1 can interact with TBK1, which is located downstream to the melanoma differentiation-associated gene 5 (MDA5) and retinoic acid-inducible gene I (RIG-I) pathways [45]. Both MDA5 and RIG-I are able to recognize the ZIKV gRNA resulting in IFN-I production. Similar results have been reported for DENV, WNV, and hepatitis C virus (HCV). In a human astrocytes cell line, ZIKV NS3 protein can act as an antagonist interacting with scaffold proteins 14-3-3ε and 14-3-3η, which are important to promote signaling of RIG-I and MDA5 through mitochondrial antiviral-signaling protein (MAVS) [46,47,48,49] (Figure 1a). Recent reports performed in microglia cells suggested that the ZIKV NS2B-NS3 complex can interact with caspases in order to inhibit apoptosis [47,50] (Figure 1a). In addition, ZIKV NS4A and NS4B cooperatively suppress the Akt-mTOR pathway, which resulted in the induction of the autophagy process and the promotion of viral replication in human fetal neural stem cells [51]. DENV NS1 protein has been involved with the activation of the immune response, interacting with some molecules of the complement system as well as TLR2/TLR6/TLR4 and STAT3, which are important for TNF-α and IL-6 production (Figure 1a) [52,53,54]. These examples suggest an active mechanism used by DENV in order to extend cell survival and promote its own replication in different cells. On the other hand, DENV NS1 induces the synthesis of antibodies that cross-react with fibrinogen, thrombocytes, and endothelial cells, resulting in endothelial dysfunction [55,56]. NS1 from ZIKV seems to have the same function modulating some molecules related to IFN-I production and inflammasome activation [57] (Figure 1a). Based on this evidence, ZIKV NS1 protein was proposed as an interesting target for the development of antiviral drugs and thus, several studies have proposed the development of neutralizing antibodies and vaccines against this viral peptide [58,59].

There has been recent progress towards the interplay between ZIKV pathogenesis and the immune system in mice models. The generation of mouse-adapted ZIKV strains together with the development of knock-in mice bearing human STAT2 have emerged as relevant tools for a better understanding of the ZIKV pathogenesis in model organisms [60]. On the other hand, mice deficient in IFNAR1 are susceptible to ZIKV infection [61] and it has been demonstrated that systemic ZIKV spreading relies on IFNAR1 expression in the hematopoietic compartment, whereas spreading within the CNS requires IFNAR1 expression on non-hematopoietic cells [62].

Innate and adaptive immunity have shown to play relevant roles in controlling ZIKV infection but in certain contexts, immune cells have also been shown to promote pathological signs of the disease, such as the increase in the expression of proinflammatory cytokines, cross-reactive antibodies as well as chronic inflammation in neural compartments [63,64]. Moreover, ZIKV replication in the brain induces marked recruitment of antiviral CD8+ T cells to the brain tissue, which limits viral replication but, in turn, promotes immunopathological features associated with the infection such as ZIKV-associated paralysis [65]. Furthermore, adaptive immunity has also been implied in controlling ZIKV infection in the testes and the brain [66].

Studies in pregnant rhesus macaques infected with ZIKV revealed that animals with prolonged viremia presented proliferation of CD16+ NK cells and CD8+ effector T cells but there were no qualitative differences with non-pregnant controls [67]. Recently, multiplex analysis of 69 cytokines was performed to explore the potential correlation between ZIKV-induced alteration of maternal immunity with fetal abnormalities, revealing that chemokines CXCL10, CCL2, and CCL8 have a high correlation with symptomatic ZIKV infection during pregnancy. In addition, the same study revealed that CCL2 presented an inverse correlation with CD163, TNFRSF1A, and CCL22 in ZIKV+ pregnant women with fetal abnormalities, with a high CCL2/(CD16/TNFRSF1A/CCL22) ratio when compared to healthy women [68]. In addition, a rare case of ZIKV infection during pregnancy in association with Guillain-Barré syndrome revealed an increase in placental inflammation and dysfunction with high cellularity (Hofbauer cells and T CD8+ lymphocytes) in tissue, and high expression of local proinflammatory cytokines such as IFN-γ and TNF-α, and other markers, such as RANTES/CCL5 and VEGFR2. [69]. A recent study analyzing immune cell profiles from ten women with confirmed ZIKV infection has shown marked changes in frequencies of monocytes, dendritic cells, plasmablasts and CD8+ T cell compartments early upon infection. These changes were followed by a rapid return to basal levels, similar to that observed in control individuals, indicating that changes in immune cell compartments are temporary [70]. The majority of ZIKV-specific CD4+ T cells were producers of TNF whereas ZIKV-specific CD8+ T cells produce TNF and IFN-γ [71]. Additional studies have also correlated changes in magnitude and frequencies of T cells upon previous exposure to DENV [72]. 

The immune response against viral components is important to control viral infection, at the same time these viral components are important to evade the host cell response for successful viral replication. However, an efficient immune response can be counter-productive because it can be exacerbated and produce tissue damage at a different scale. The temporality of viremia is also crucial because the virus has different tropisms and the immune response needs to deal with this temporality. The interplay between the immune response and viral components is important to obtain a comprehensive understanding of ZIKV infection and the scope of innate and adaptive immunity [63,64].

## 4. ZIKV Remodels the ER and Induces ER Stress

As mentioned above, ZIKV replication and budding at the ER lumen occurs inside viral factories, which are formed by convoluted membrane rearrangements generated from an increase in the local membrane production. The exact mechanism that allows the assembly of this structure is not completely understood, but it has been determined that it is induced by the ZIKV NS1 protein [73]. This local change in the ER membrane is only observed in mammalian models of flaviviruses infection as it is not observed in mosquito-derived cell lines such as C6/36 [74]. A possible explanation for this is the differential metabolic changes that undergo mosquito and human cell lines upon ZIKV infection where an increment in glucose utilization through the pentose phosphate pathway was observed in mosquito cells while glucose is preferentially used in the tricarboxylic acid cycle in human cells [75]. In the same line, infections in human microglia lead to modulation in the synthesis of lysophospholipids, phospholipids, and carboxylic acids that are involved in membrane structure and viral replication [76].

The physical and metabolic changes that occur at the ER upon ZIKV infection lead to the activation of the unfolded protein response (UPR) [77,78]. The UPR is a conserved three-pronged signaling pathway that senses the presence of unfolded or misfolded proteins in the ER lumen and is aimed to preserve ER homeostasis through the inhibition of protein synthesis and the transcriptional induction of genes coding for chaperones and genes associated to a protein degradation pathway known as ER-associated degradation (ERAD), among others [79]. The UPR has three main sensors, protein kinase R-like ER-resident kinase (PERK), activating transcription factor 6 (ATF6), and the inositol-requiring enzyme 1 (two homologs: IRE1α present in most tissues and IRE1β, which is specific in intestinal epithelium) (Figure 1b). These receptors are located in the ER membrane with the sensory domain pointing towards the ER lumen, while the kinase and endonuclease domain faces the cytoplasm, both linked by a transmembrane domain that has also been involved in ER lipid composition sensing [80]. The sensory domain of the three receptors is associated with the ER chaperone binding immunoglobulin protein, BiP (also known as Grp78), preventing their aggregation and activation [81] (Figure 1b).

PERK activation begins with the dissociation of BiP and is followed by dimerization and autophosphorylation of the C-terminal kinase domain. In its active state, PERK phosphorylates the eukaryotic translation initiation factor 2α (eIF2α), which leads to an attenuation of global protein synthesis. However, some mRNA carrying internal ribosome entry sites (IRES) such as those coding for ribosomal proteins or the transcription factor ATF4, are preferentially translated during eIF2α phosphorylation. ATF4 is a transcription factor that drives the expression of pro-survival genes involved in the amino acid synthesis and transport but can also induce the synthesis of pro-apoptotic factors such as C/EBP homologous protein (CHOP), thus triggering apoptosis [82] (Figure 1b). Pharmacological intervention with a PERK inhibitor in a mouse model of ZIKV infection (IFNɑ/β receptor null mice infected intracerebroventricularly) restored neurogenesis and resulted in an impairment of microcephaly development without affecting viral replication, indicating that PERK activation is involved in ZIKV pathogenesis in the brain but not in viral replication in vivo [83].

On the other hand, upon dissociation of BiP from the ATF6 N-terminal domain, the receptor undergoes translocation to the Golgi apparatus followed by intramembrane proteolysis by S1P and S2P proteases. This proteolytic processing of ATF6 leads to the release of the N-terminal domain of ATF6, which acts as a transcription factor. As such, ATF6 translocates to the nucleus inducing expression of specific genes including BiP, X box-binding protein 1 (XBP1,) and components of the ERAD machinery [84] (Figure 1b). However, compared with PERK and IRE1 pathways, ATF6 has remained understudied and thus, more studies on the role of ATF6 during ZIKV infection are needed to better understand the role of this ER stress-related pathway on viral replication and pathogenesis.

The pathway leading to IRE1α activation upon dissociation of BiP is similar to that of PERK. IRE1α dimerization allows the autophosphorylation of the C-terminal kinase domain, which turns on the endoribonuclease activity of the receptor. Once activated, IRE1α performs a non-canonical splicing event that removes a 26-nucleotide intron present in the XBP1 mRNA generating a frameshift that allows the synthesis of a full XBP1 protein (XBP1s), which has transcriptional regulatory activity [85] (Figure 1b). XBP1s act as a homodimer regulating the expression of ER chaperones and components of the ERAD machinery [86]. However, the hyperactivation of IRE1α can lead to its oligomerization and an increase in the endoribonuclease activity, which triggers an RNA decay pathway known as Regulated Ire1-Dependent Decay (RIDD) [87]. During RIDD, IRE1α cleaves RNA targets preferentially localized at the ER that contain a consensus sequence similar to the cleavage site present in the XBP1 mRNA [88]. Several RIDD targets have been described in mammals and a few specifically in human cells, but there is still uncertainty whether this RNA degradation mechanism has a regulatory role in particular pathways or if it only has a role in the downregulation of the protein synthesis by reducing mRNA levels [89,90,91,92]. Alternatively, active IRE1α can recruit the tumor necrosis factor receptor-associated factor 2 (TRAF2) and induce signal-regulating kinase (ASK1), cJUN NH2-terminal kinase (JNK), and p38MAPK activation. The final activation of the JNK kinase leads to the activation of pro-apoptotic proteins such as Bim as well as the induction of apoptosis through caspase 3 [93] (Figure 1b). In contrast with what was observed with the PERK inhibitor, which decreased microcephaly without interfering with viral replication, treatment with the IRE1α inhibitor 4μ8C resulted in decreased ZIKV replication in the brain and prevented microcephaly in a IFNɑ/β receptor null mouse model, indicating the critical role of IRE1α RNase activity in ZIKV replication and pathogenesis [83]. However, it is still unknown whether only IRE1α-dependent XBP1 mRNA splicing and/or RIDD are required for ZIKV replication and pathogenesis in brain tissue.

Given the central role of the ER and the UPR in maintaining cellular homeostasis, it is not surprising that infection with other ER-replicating flaviviruses activates the different branches of this pathway. Indeed, DENV activates all arms of the UPR sensory pathway in mammalian epithelial and fibroblast cell lines [94]. Reports in human fibrosarcoma cells showed UPR activation in a time-dependent manner; with PERK induced at early stages of infection, followed by the IRE1α-XBP1 axis and, finally, ATF6 in mid and late replication cycle stages [95]. It was proposed that DENV prM, E and NS1 glycoproteins are responsible for inducing the UPR pathway in adenocarcinoma cell lines, but the pattern of activation was determined by the viral serotype [96]. In general, UPR activation is detrimental for DENV replication; however, the XBP1 function was shown to be beneficial for DENV replication and plays a role in reducing stress-induced cell cytotoxicity [97]. In the case of WNV, studies in a mammalian kidney cell line suggest that the viral protein NS4 is able to induce XBP1 mRNA splicing through an unknown mechanism. This would interfere with the JAK/STAT signaling during infection thus, facilitating replication and immune response evasion [98]. On the other hand, infection in neuroblastoma cells leads to high levels of ATF6 activation, with transient eIF2α phosphorylation, while XBP1 splicing seems to have no role in viral replication. In this case, WNV-induced ATF6 and PERK signaling lead to activation of CHOP-dependent apoptosis, explaining the high neuropathogenesis of the virus [99]. Moreover, JEV is also able to induce XBP1 mRNA splicing and CHOP induction, while RIDD activity was shown to be beneficial for viral replication [100]. Interestingly, a recent report showed that tick-borne encephalitis virus (TBEV) infection induces UPR in U2OS cells through ATF6, PERK, and IRE1α activation but only IRE1α showed an antiviral property. The authors also reported that this effect was conserved in several flaviviruses including DENV, WNV, and ZIKV and was dependent on the JNK/IRF3 pathway but independent of the IRE1α RNAse activity. The early induction of UPR resulted in the priming of the innate immune response allowing the cell to mount a potent antiviral response resulting in lower viral production [101]. 

Hepatitis C virus (HCV), a more distant member of the Flaviviridae family (Hepacivirus genus), also modulates the three UPR branches with a tendency to block the activity of the sensors resulting in IRE1α-XBP1 suppression [102], ATF6 cleavage but CHOP inhibition [103] and PERK repression through interaction with the viral protein E1 [104]. Structural proteins C and E from HCV are the main factors activating the UPR response [105,106], while non-structural proteins play a regulatory role in vitro and in vivo [107].

Some studies have been carried out in order to understand the impact of ZIKV infection in the UPR response, mostly in neural tissue. So far, ZIKV seems to activate the three UPR branches although this effect is dependent on the cell type [77,78]. According to the latest reports, ZIKV infection significantly upregulates phosphorylated IRE1α and XBP1 mRNA splicing as well as ATF6 activation in mice cerebellum, indicating a tissue dependency in the UPR activation profile [77]. Something similar was observed in whole brain tissue when infecting IFN KO mice in a vertical transmission model, where XBP1 and ATF6 activation was correlated with a microcephaly-like phenotype, and pharmacological inactivation of the UPR alleviated the pathological effects of ZIKV [83]. Although ZIKV upregulates PERK activity, it does not significantly increase ATF4 activation, possibly by an active regulation of eIF2α dephosphorylation, a strategy used by the virus to decrease stress granules formation and promote its own gene expression [108] (see below). On the other hand, studies performed in a human alveolar epithelial cell line (A549) lead to the conclusion that viral infection leads to ATF6 inhibition through an unknown mechanism [109] (Figure 1a). This is not particularly surprising considering that previous transcriptomic analysis revealed a significant difference in gene expression among infected human cell lines, even when comparing functionally similar cells like THP1-derived macrophages and macrophage-like microglia [25]. Therefore, it is important to define what ER stress means when extrapolating data from infection assays in tissue and whole organisms, and how the profiles of UPR-regulated genes could lead to the observed pathologies in clinical subjects. So far, few studies have been done on human brain-derived cell lines with highly pathogenic strains of ZIKV, especially those isolated in South America, and although the exact mechanism of virally induced UPR modulation is still unclear, the general consensus is that UPR activation upon ZIKV infection is a major contributor to viral pathogenesis [78] (Figure 1b).

## 5. Regulation of Viral and Host Cell Gene Expression during ZIKV Replication

Following viral infection, the infected cell responds by mounting an integral component of the innate immune response known as an integrated stress response (ISR). ISR involves the shut-off of cellular protein synthesis mediated by the phosphorylation of eukaryotic translation initiation factor 2 alpha (eIF2α) by one or more of four members of the eIF2α kinase family (PERK, PKR, HRI, and GCN2) [110]. These events can be accompanied by the induction of RNA granules assembly, expression of selected genes important to promote cellular recovery (Activating transcription factor ATF4, X-box binding protein 1 XBP1) and/or the activation of the unfolded protein response (UPR) [110].

Despite that it contains a cap0 and cap1 structure, it was shown that the 5´-untranslated region (5´-UTR) of the DENV and ZIKV gRNA has the ability to recruit ribosomes through a cap-independent, internal ribosome entry site (IRES)-driven mechanism both in mammalian (BHK) and mosquito (C6/36) cells [111,112]. Consistent with the use of an alternative mechanism of translation initiation, it was reported that replication of these two flaviviruses induces eIF2α phosphorylation resulting in a potent blockade of host protein synthesis in Huh7 cells [113,114]. Interestingly, despite this potent induction of eIF2α phosphorylation and inhibition of host cell protein synthesis, it was shown that ZIKV replication inhibits stress granules assembly in Huh7, U2OS, and A549 cells as well as in human fetal astrocytes [108,113,114,115]. While host cell protein synthesis was shown to be inhibited by ZIKV NS3 and NS4A, stress granules assembly was interfered with by the action of the capsid, NS3/NS2B, and NS4A [114]. Alternatively, it was shown that ZIKV subverts stress granules assembly by sequestering stress granules dependency factors such as G3BP1 towards viral replication sites [114,115]. The mechanism of subversion of stress granules assembly by sequestering dependency factors has also been shown for other members of the flavivirus genus including DENV, WNV, and JEV [116,117].

ZIKV infection also influences host cell transcriptional landscapes. Indeed, transcriptome-wide analyses of human microglia, fibroblasts, embryonic kidney, and monocyte-derived macrophage cell lines, as well as neuronal stem cells, showed that ZIKV infection induced a transcriptional reprogramming [25,118]. Interestingly, analysis of host mRNA and miRNA interaction networks of human neural stem cells infected by ZIKV revealed miRNA-mediated regulation of different cellular processes such as cell cycle, stem cell maintenance, and neurogenesis. These analyses also revealed several miRNA withpredicted involvement in the development of microcephaly pointing out to an important role of ZIKV-induced transcriptional reprogramming in viral pathogenesis [118].

## 6. Interplay between RNA Modifications, ZIKV Replication and Cellular Responses

RNA molecules from bacteria, archaea, and eukaryotes contain more than 100 chemical modifications [119]. While most of these modifications have been described in non-coding RNA such as tRNA and rRNA, some of them have also been identified in mRNA and viral RNA molecules [120]. Interestingly, affinity-purified viral RNA from Huh7-infected cells and viral particles of HCV, DENV, and ZIKV were shown to contain several chemical modifications including dimethylcytosine species (m^5^Cm and m^4^_4_C) and N^6^-methyladenosine (m^6^A) [121]. In this regard, m^6^A is the most abundant internal modification described in Eukaryotic mRNA and viral RNA. It is deposited co-transcriptionally in the nucleus by the writer complex mainly composed of methyltransferases METTL3 and METTL14 together with the cofactors WTAP, KIAA1429 amongst others [122]. Deposition of m^6^A occurs mainly at the DRACH consensus sequence and has been shown to be reversed by two RNA demethylases, FTO and ALKBH5 [122]. The effect of m^6^A in RNA metabolism is exerted by m^6^A-binding proteins from which members of the YTH domain-containing family of proteins have been the better characterized [123]. Depending on the reader protein that recognizes the methylated mRNA and the position of the m^6^A residue along the body of the mRNA, this modification can modulate alternative splicing (YTHDC1), nuclear export (YTHDC1), cytoplasmic localization (YTHDF1, 2, and 3), translation (YTHDF1, YTHDF3, YTHDC2), and decay (YTHDF2, YTHDF3) [122]. 

The RNA of ZIKV and other members of the *Flaviviridae* family were shown to be decorated with m^6^A [124,125]. Interestingly, these reports showed some differences in the methylation patterns between Asian and African lineages of ZIKV but also between cell types (HEK293T or Huh7 cells) [124,125], suggesting that site- and cell-dependent methylation of the viral RNA may play an important role in viral replication and pathogenesis. From a functional point of view, it was shown that the presence of m^6^A in the ZIKV gRNA plays a negative role during viral replication in HEK293T cells [124]. As such, knockdown of RNA methyltransferases METTL3 or METTL14 was associated with an increase in viral protein levels, viral RNA production and, as a consequence, viral titers [124]. As expected, depletion of the m^6^A demethylase ALKBH5 induced the opposite effects [124]. The negative effect on viral replication was exerted by the m^6^A reader proteins YTHDF1, YTHDF2, and YTHDF3, which bind the viral RNA in an m^6^A-dependent fashion and exert a negative role on ZIKV replication [124] (Figure 1a). Although it was shown that the YTHDF2 protein, which was previously associated with accelerated degradation of its methylated mRNA targets [126], was the most potent inhibitor of viral replication, the mechanism by which YTHDF proteins exert a rather redundant role was not further explored. A similar redundant, negative effect of YTHDF proteins was shown during HCV replication [125]. Here, YTHDF proteins were relocalized to viral replication sites at lipid droplets during HCV replication. Interestingly, the same study showed that the viral core protein binds preferentially to viral RNA that lacks m^6^A at the E1 region suggesting that the virus avoids the incorporation of methylated RNA into newly produced particles [125]. Whether m^6^A reader proteins are relocalized to the ZIKV assembly sites at the ER during viral replication or whether ZIKV capsid protein packages hypomethylated RNA is currently unknown (Figure 1a). In addition, m^6^A-mediated regulation has been shown to be very active in the brain and thus, it would be of interest to study the role of m^6^A and its associated machinery on ZIKV replication in brain-derived cell types.

Although Lichinchi and colleagues suggested that ZIKV replication induced changes in the m^6^A patterns of HEK293T mRNA [124], these observations were recently challenged [127]. By performing an unbiased bioinformatic protocol to detect m^6^A sites using MeRIP-seq data [128], Gokhale and colleagues identified and validated changes in m^6^A sites in Huh7 cells infected with ZIKV, DENV, WNV, and HCV [127]. The authors showed that infection with these members of the *Flaviviridae* family resulted in altered patterns of m^6^A in specific cellular transcripts. However, while several changes were virus-specific, the authors were able to identify conserved alterations. Amongst the transcripts with altered m^6^A patterns that were identified with all viruses tested are those coding for RIOK3 and CIRBP, which presented changes dependent on the innate immune sensing and ER stress pathways, respectively. As such, in addition to infection, the appearance of m^6^A sites in the RIOK3 mRNA was also induced in cells treated with the HCV PAMP or IFN-β but was not observed in IRF3 KO cells, indicating that innate immune signaling pathway was involved in the induction of RIOK3 mRNA methylation. On the other hand, the loss of an m^6^A site in the CIRBP mRNA was induced by a viral infection and the ER stress inducer drug thapsigargin. However, no changes in the methylation pattern of CIRBP mRNA were observed in IRF3 KO cells or in cells treated with HCV PAMP or IFN-β, indicating that this change was specific to ER stress. While the addition of m^6^A to the RIOK3 mRNA was shown to increase its translational rates, the loss of an m^6^A site in the CIRBP mRNA resulted in an altered splicing pattern. Interestingly, knockdown of RIOK3 or CIRBP proteins was associated with reduced viral titers in DENV and ZIKV infected cells suggesting that these proteins have proviral functions. The authors also validated the role of the protein products from additional 20 cellular mRNA presenting conserved changes in their m^6^A patterns in response to infection and found that most of them were able to either promote or reduce viral replication at least with one of the viruses tested [127]. These data strongly indicate that the cellular m^6^A machinery act in concert with the innate immune signaling and the ER stress pathways as a response to viral infection. Interestingly, it was recently reported that early activation of the UPR pathway during TBEV, DENV, WNV, and ZIKV results in the priming of a potent innate immune response and a cell-intrinsic inhibition of viral replication in an IRF3-dependent manner [101]. Whether this ER stress-induced priming of the innate immune response is regulated by m^6^A in response to viral infection remains to be explored. Interestingly, it was recently reported that the presence of m^6^A within the IFN-β mRNA was associated with reduced stability of this transcript and thus, to an attenuated expression of interferon-stimulated genes and an innate immune response [129,130]. In addition, m^6^A deposition within the STAT1 mRNA was associated with increased stability of the transcript and thus, increased STAT1 protein levels, which in turn promoted M1 polarization of mouse macrophages [131]. Whether an m^6^A-mediated regulation of these and other immune response genes occurs during ZIKV replication warrants further investigation.

Localization of mRNA to RNA granules such as stress granules and p-bodies is also regulated by the presence of m^6^A [132,133]. However, it has never been evaluated whether changes in the subcellular localization of cellular mRNA in response to viral infection is regulated by m^6^A and/or its associated machinery.

The NS5 protein of ZIKV, as well as that of other members of the family, contains an RNA methyltransferase domain, which performs the N^7^ methylation at the guanosine residue to form the cap structure (cap0) and the 2´-O-methylation of the ribose at the first transcribed adenosine residue (cap1). Cap methylation not only confers stability and translational properties but also allows the viral RNA to mimic cellular mRNA thus, avoiding its recognition as non-self by host RNA sensors such as MDA5, RIG-I, and IFIT1 [134]. Whether the NS5 protein of ZIKV or other members of the family is able to methylate host cell RNAs and the consequences of this (if any) have not been yet described.

## 7. Concluding Remarks

The recent outbreak of ZIKV in the Americas during 2015–2016 and its association with neurological disorders such as Guillain-Barrré syndrome and congenital microcephaly led the WHO to declare this virus as a public health emergency concern. Certainly, the human to human transmission of the virus together with its teratogenic properties made ZIKV a major threat to the human population. ZIKV is able to penetrate the placental barrier at different gestational ages and reach the brain of the fetus. Indeed, ZIKV is highly neurotropic and the virus has been shown to replicate efficiently in many different brain cells such as neural cells isolated from primary tissue, human induced pluripotent stem cell-derived neural progenitor cells, neurospheres, brain organoids but also in microglia, which are the sentinels of the brain but have a hematopoietic origin. Thus, ZIKV is quite flexible and adapts very well to replicate in different human tissues. This idea is further supported by the ability of ZIKV to infect and replicate in disparate hosts such as mosquitoes and humans. Therefore, the understanding of the specific interactions between ZIKV and its different host cells is critical to improving our knowledge of the biology of viral replication as well as on specific cellular responses to the infection. As such, the study of the crosstalk between viral and cellular RNA metabolism with cellular responses such as the innate immune response, the integrated stress response, and the unfolded protein response is critical to understand how different cells from different tissues respond to counteract or tolerate ZIKV infection (Figure 1). This knowledge, together with the use of recently developed mouse models, will be useful for the development of innovative antiviral therapies aimed to counteract the neurological disorders induced by ZIKV.

## Figures and Tables

**Figure 1 pathogens-09-00158-f001:**
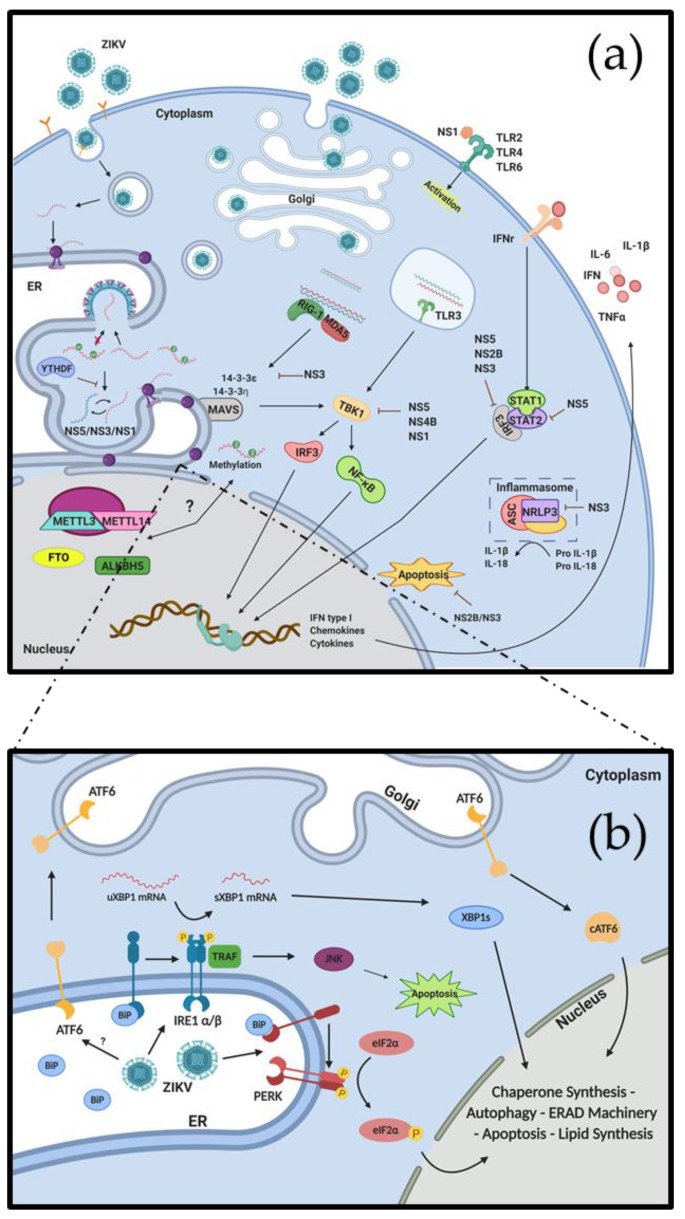
Crosstalk between RNA metabolism and cellular stress responses during Zika virus replication. ZIKV infection induces cellular stress that triggers different responses based on changes in RNA metabolism including the activation of innate immune response-related genes or changes in the methylation of host mRNA (**a**). In addition, viral replication at the endoplasmic reticulum (ER) generates ER-stress with the consequent triggering of the unfolded protein response (UPR) pathway (**b**).

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
