# Peer review of "Crosstalk between RNA Metabolism and Cellular Stress Responses during Zika Virus Replication"

_pathogens, 2020, doi:10.3390/pathogens9030158_

Round 1

Reviewer 1 Report

The review by Oyarzun-Arrau et al. in their paper "Crosstalk between RNA metabolism and cellular stress responses during Zika virus replication" is an excellent review of the subject and extends the knowledge base of Zika virus pathogensis. The paper is well researched and draws important correlations across what has become a vast literature base.

There are some minor comments and critiques for the authors to consider

As the authors discuss the role of the NS proteins in pathogenesis, papers regarding NS4A/B on neural disease and autophagy may have relevance, for example the paper by Liang et al (PMID:27524440). The section describing animal models (Lines 155-177) can be significantly contracted. While interesting, the focus on cell mechanisms should be maintained and this information is largely superfluous. The authors should perhaps separate out host-responses against NS proteins as a means to control and limit morbidity of disease in a separate section, however, such information needs to be more complete and perhaps left to a separate monograph. The paragraph starting on Line 178 does not flow well with the remainder of the paragraph. In lines 178-183 the authors discuss the observation that leukocyte populations are altered early in infection, however, how this relates to either neurologic disease (the immediate preceding paragraph or immune responses to infection (the central focus of this section) are not clear. Unless relevance can be established, this can be removed.

4. In the same paragraph, lines 183-188 focus on the type (IgG, IgM, IgA) antibody responses against Zika. Again, relevance to the central theme of the section is not clear. These lines could also be deleted without harming the section. 

5. In the same paragraph, the last portion in lines 188-196 describes the human immune response to infection. While immune activation against the NS proteins would be important - the authors need to better link such observations to clinical outcome.

Author Response

Dear Reviewer, 

Thank you very much for your comments, which have largely help us to improve the quality of our manuscript.

Please find below a point-by-point reply to your comments:

The review by Oyarzun-Arrau et al. in their paper "Crosstalk between RNA metabolism and cellular stress responses during Zika virus replication" is an excellent review of the subject and extends the knowledge base of Zika virus pathogensis. The paper is well researched and draws important correlations across what has become a vast literature base.

Thank you very much!

There are some minor comments and critiques for the authors to consider

As the authors discuss the role of the NS proteins in pathogenesis, papers regarding NS4A/B on neural disease and autophagy may have relevance, for example the paper by Liang et al (PMID:27524440). The section describing animal models (Lines 155-177) can be significantly contracted. While interesting, the focus on cell mechanisms should be maintained and this information is largely superfluous. The authors should perhaps separate out host-responses against NS proteins as a means to control and limit morbidity of disease in a separate section, however, such information needs to be more complete and perhaps left to a separate monograph. The paragraph starting on Line 178 does not flow well with the remainder of the paragraph. In lines 178-183 the authors discuss the observation that leukocyte populations are altered early in infection, however, how this relates to either neurologic disease (the immediate preceding paragraph or immune responses to infection (the central focus of this section) are not clear. Unless relevance can be established, this can be removed.

The following sentence and the corresponding reference were included in the revised version:

“In addition, ZIKV NS4A and NS4B cooperatively suppress the Akt-mTOR pathway, which resulted in the induction of the autophagy process and the promotion of viral replication in human fetal neural stem cells [51]”.

As suggested by the Reviewer, the section describing the animal models was shortened and we have also included a separated paragraph on the host-response against NS proteins, describing some examples. Finally, the observation regarding leukocytes was removed as suggested.

  1. In the same paragraph, lines 183-188 focus on the type (IgG, IgM, IgA) antibody responses against Zika. Again, relevance to the central theme of the section is not clear. These lines could also be deleted without harming the section. 

This section was removed as suggested

  1. In the same paragraph, the last portion in lines 188-196 describes the human immune response to infection. While immune activation against the NS proteins would be important - the authors need to better link such observations to clinical outcome.

This section was paraphrased discussing the importance of the immune response against NS proteins

Reviewer 2 Report

REVIEW

Dear Author,

The review manuscript summarizes relevant aspects of the complex crosstalk between RNA metabolism and cellular stress responses against ZIKV and discusses their possible impact on viral pathogenesis.

Overall:  The manuscript is fairly well written but there are several statements that need to be revised for clarity.  The paper disregards the current state of ZIKV infection after 2016, ZIKV pathogenesis in the ocular compartment, cell types that are permissive in the human kidney for ZIKV, as well the immune response in mothers infected with ZIKV and their babies.

My final analysis is that the manuscript be revised based on my comments.

Major Comments

1.In the pathogenesis overview, there is no mentioning of cell types infected in the human ocular compartment (retinal pericytes, retinal endothelial, and retinal pigmented epithelial cells etc.) as well as the severe ocular pathology found in infants born to ZIKV infected mother.

2.  In the pathogenesis overview, there no mentioning of cells permissive for ZIKV infection in glomerulus of the human kidney (podocytes, mesangial cells, glomerular endothelial cells.

3. In the pathogenesis overview there is no mentioning of the current global state of ZIKV infection after 2016. Where is ZIKV infection in 2020?

4. There are many acronyms in the Figures 1a and 1b that should be spelled out for the reader in the legend, text, or separately.

5. There is no information in the manuscript that examines immune responses in mother infected with ZIKV or their babies.

6. There are very few references for the reader to go back to the figure.  

Minor comments

1. Line #77 poliA tail at the 3’ end (it should be “poly A” tail)

2. Line 83 “AXL, Tyro3 or TIM1” there is controversy surrounding these receptors

3.  Figure 1a the word “cytoplasm” is misspelled.

4. Line #139 The specific type of interferon induced by ZIKV should be stated.

5.  Line #145 should say in “different cells” and not cell lines.

6. Line 147 should say “complement system” and not the complement.

7. Line #158 omit the word “the”   

8. Line #175 “limits viral replication but it turn” needs to be modified.

9.  Lines 204 and 205 “metabolic changes that suffer mosquito versus human cell line” is poorly worded and needs to be modified.

10. Line #205 “where an increment in glucose” needs to be revised.

11. Line #316 especially those isolated in the “North” American continent.

12. Line #362 “Depending “of” the reader protein” (on).

13.  Line #374 and 374 “protein, released viral RNA and viral titers” is poorly worded.

14. Line 382-384, there is no reference for this observation “Interestingly, Gokhale and colleagues showed that the viral core protein binds preferentially to viral RNA that lacks m6A at the E1 region suggesting that the virus avoids the incorporation of methylated RNA into newly produced particles.”

Author Response

Dear Reviewer, 

Thank you very much for your comments, which have largely help us to improve the quality of our manuscript.

Please, find below a point-by-point reply to your comments.

Dear Author,

The review manuscript summarizes relevant aspects of the complex crosstalk between RNA metabolism and cellular stress responses against ZIKV and discusses their possible impact on viral pathogenesis.

Thank you very much

Overall:  The manuscript is fairly well written but there are several statements that need to be revised for clarity.  The paper disregards the current state of ZIKV infection after 2016, ZIKV pathogenesis in the ocular compartment, cell types that are permissive in the human kidney for ZIKV, as well the immune response in mothers infected with ZIKV and their babies.

My final analysis is that the manuscript be revised based on my comments.

Major Comments 

1.In the pathogenesis overview, there is no mentioning of cell types infected in the human ocular compartment (retinal pericytes, retinal endothelial, and retinal pigmented epithelial cells etc.) as well as the severe ocular pathology found in infants born to ZIKV infected mother.

We have included the following paragraph with the corresponding references in the revised version:

ZIKV is also able to infect blood-retinal barrier cells like retinal endothelial cells, retinal pericytes, and retinal pigmented epithelial cells leading to damage in the retina, optic nerve, and retinal vessels in 20 to 55% of congenital ZIKV infections and, although infrequent, major visual impairment in non-congenital infections [16,17].

  1. In the pathogenesis overview, there no mentioning of cells permissive for ZIKV infection in glomerulus of the human kidney (podocytes, mesangial cells, glomerular endothelial cells.

We have included the following paragraph with the corresponding references in the revised version:

There is also evidence showing that ZIKV can infect human glomerular cells such as human podocytes, renal glomerular endothelial cells and mesangial cells [19]. Despite only a couple of clinical cases of ZIKV-associated renal complications have been observed [20], experimental observations suggest that ZIKV could induce acute kidney injury through Nod-like receptor protein 3 (NLRP3) inflammasome activation through the suppression of BCL-2 expression [21].

  1. In the pathogenesis overview there is no mentioning of the current global state of ZIKV infection after 2016. Where is ZIKV infection in 2020?

We have included the following paragraph with the corresponding references in the revised version:

Although ZIKV infections have substantially declined since 2016, as of July 2019 several countries in the Americas reported cases of ZIKV transmission, with the exception of mainland Chile, Uruguay and Canada. Moreover, by 2018 evidence indicated that ZIKV strains found in the Americas had spread to Angola and were associated with a cluster of microcephaly cases. [4, 5].

  1. There are many acronyms in the Figures 1a and 1b that should be spelled out for the reader in the legend, text, or separately.

 Acronyms of Figure 1 were included in the Figure legend

  1. There is no information in the manuscript that examines immune responses in mother infected with ZIKV or their babies.

We have included the following paragraph with the corresponding references in the revised version:

Studies in pregnant rhesus macaques infected with ZIKV revealed that animals with prolonged viremia presented proliferation of CD16+ NK cells and CD8+ effector T cells but there were no qualitative differences with non-pregnant controls [67]. Recently, a multiplex analysis of 69 cytokines was performed to explore the potential correlation between ZIKV-induced alteration of maternal immunity with fetal abnormalities, revealing that chemokines CXCL10, CCL2, and CCL8 have a high correlation with symptomatic ZIKV infection during pregnancy. In addition, the same study revealed that CCL2 presented an inverse correlation with CD163, TNFRSF1A, and CCL22 in ZIKV+ pregnant women with fetal abnormalities, with a high CCL2/(CD16/TNFRSF1A/CCL22) ratio when compared to healthy women [68]. In addition, a rare case of ZIKV infection during pregnancy in association with Guillian-Barré syndrome revealed an increase in placental inflammation and dysfunction with high cellularity (Hofbauer cells and T CD8+ lymphocytes) in tissue, and high expression of local proinflammatory cytokines such as IFN-γ and TNF-α, and other markers, such as RANTES/CCL5 and VEGFR2. [69]. A recent study analyzing immune cell profiles from ten women with confirmed ZIKV infection has shown marked changes in frequencies of monocytes, dendritic cells, plasmablasts and CD8+ T cell compartments early upon infection. These changes were followed by a rapid return to basal levels, similar to that observed in control individuals, indicating that changes in immune cell compartments are temporary [70]. The majority of ZIKV-specific CD4+ T cells were producers of TNF whereas ZIKV-specific CD8+ T cells produce TNF and IFN-γ [71]. Additional studies have also correlated changes in magnitude and frequencies of T cells upon previous exposure to DENV [72].

  1. There are very few references for the reader to go back to the figure.  

 Figure 1a and b were referenced along the text when pertinent

Minor comments

  1. Line #77 poliA tail at the 3’ end (it should be “poly A” tail)

 Corrected accordingly

  1. Line 83 “AXL, Tyro3 or TIM1” there is controversy surrounding these receptors

The section was modified as follows in the revised version:

The viral replication cycle begins with the attachment of the envelope protein E to tyrosine kinase receptors such as AXL, Tyro3 or TIM1, leading to virion endocytosis by a clathrin-dependent mechanism [27]. Nevertheless, there is controversy whether these receptors are indeed essential for viral entry [28].

  1. Figure 1a the word “cytoplasm” is misspelled.

 Corrected accordingly

  1. Line #139 The specific type of interferon induced by ZIKV should be stated.

The specific type of interferon was stated in every section were this molecule was mentioned.

  1. Line #145 should say in “different cells” and not cell lines.

 Corrected accordingly

  1. Line 147 should say “complement system” and not the complement.

 Corrected accordingly

  1. Line #158 omit the word “the”   

Corrected accordingly

  1. Line #175 “limits viral replication but it turn” needs to be modified.

 Corrected accordingly

  1. Lines 204 and 205 “metabolic changes that suffer mosquito versus human cell line” is poorly worded and needs to be modified.

We have modified the paragraph in the revised version as follows:

“A possible explanation for this is the differential metabolic changes that undergo mosquito and human cell lines upon ZIKV infection, where an increment in glucose utilization through the pentose phosphate pathway was observed in mosquito cells while glucose is preferentially used in the tricarboxylic acid cycle in human cells [75]. In the same line, infections in human microglia lead to a modulation in the synthesis of lysophospholipids, phospholipids and carboxylic acids that are involved in membrane structure and viral replication [76].

  1. Line #316 especially those isolated in the “North” American continent.”   In Lines# 209 to 215

 Corrected accordingly

  1. Line #362 “Depending “of” the reader protein” (on).

Corrected accordingly

  1. Line #374 and 374 “protein, released viral RNA and viral titers” is poorly worded.

 We have modified the paragraph in the revised version as follows:

“As such, knockdown of RNA methyltransferases METTL3 or METTL14 was associated with an increase in viral protein levels, viral RNA production and, as a consequence, viral titers [124]. As expected, depletion of the m6A demethylase ALKBH5 induced the opposite effects [124].”

  1. Line 382-384, there is no reference for this observation “Interestingly, Gokhale and colleagues showed that the viral core protein binds preferentially to viral RNA that lacks m6A at the E1 region suggesting that the virus avoids the incorporation of methylated RNA into newly produced particles.”

The observation was linked to the corresponding reference as follows:

“Interestingly, the same study showed that the viral core protein binds preferentially to viral RNA that lacks m6A at the E1 region suggesting that the virus avoids the incorporation of methylated RNA into newly produced particles [125].”

Round 2

Reviewer 2 Report

REVIEW

Dear Author,

The review manuscript summarizes relevant aspects of the complex crosstalk between RNA metabolism and cellular stress responses against ZIKV and discusses their possible impact on viral pathogenesis.

Overall:  The manuscript is fairly well written and the authors have addressed my major and minor concerns in the revised manuscript.